# Goal Attainment in an Individually Tailored and Home-Based Intervention in the Chronic Phase after Traumatic Brain Injury

**DOI:** 10.3390/jcm11040958

**Published:** 2022-02-12

**Authors:** Ida M. H. Borgen, Solveig L. Hauger, Marit V. Forslund, Ingerid Kleffelgård, Cathrine Brunborg, Nada Andelic, Unni Sveen, Helene L. Søberg, Solrun Sigurdardottir, Cecilie Røe, Marianne Løvstad

**Affiliations:** 1Department of Physical Medicine and Rehabilitation, Oslo University Hospital, 0424 Oslo, Norway; mavfor@ous-hf.no (M.V.F.); uxinff@ous-hf.no (I.K.); nada.andelic@medisin.uio.no (N.A.); unsvee@ous-hf.no (U.S.); h.l.soberg@medisin.uio.no (H.L.S.); cecilie.roe@medisin.uio.no (C.R.); 2Department of Psychology, Faculty of Social Sciences, University of Oslo, 0316 Oslo, Norway; solveigl@ous-hf.no (S.L.H.); marianne.lovstad@sunnaas.no (M.L.); 3Department of Research, Sunnaas Rehabilitation Hospital, 1453 Nesoddtangen, Norway; 4Oslo Centre for Biostatistics and Epidemiology, Oslo University Hospital, 0424 Oslo, Norway; uxbruc@ous-hf.no; 5Center for Habilitation and Rehabilitation Models and Services (CHARM), Institute of Health and Society, Faculty of Medicine, University of Oslo, 0316 Oslo, Norway; 6Department for Occupational Therapy Prosthetics and Orthotics, Faculty of Health Sciences, Oslo Metropolitan University, 0130 Oslo, Norway; 7Department of Physiotherapy, Faculty of Health Sciences, Oslo Metropolitan University, 0130 Oslo, Norway; 8Centre for Rare Disorders, Oslo University Hospital, 0424 Oslo, Norway; sosigu@ous-hf.no; 9Institute of Clinical Medicine, Faculty of Medicine, University of Oslo, 0316 Oslo, Norway

**Keywords:** traumatic brain injury, goal-oriented rehabilitation, home-based rehabilitation, community-based rehabilitation, SMART, goal attainment scaling (GAS)

## Abstract

Traumatic brain injury (TBI) is a heterogeneous condition with long-term consequences for individuals and families. Goal-oriented rehabilitation is often applied, but there is scarce knowledge regarding types of goals and goal attainment. This study describes goal attainment in persons in the chronic phase of TBI who have received an individualized, SMART goal-oriented and home-based intervention, compares goal attainment in different functional domains, and examines indicators of goal attainment. Goal attainment scaling (GAS) was recorded in the intervention group (*n* = 59) at the final session. The goal attainment was high, with 93.3% increased goal attainment across all goals at the final session. The level of goal attainment was comparable across domains (cognitive, physical/somatic, emotional, social). Gender, anxiety symptoms, self-reported executive dysfunction, and therapy expectations were indicators of goal attainment. These results indicate a potential for the high level of goal attainment in the chronic phase of TBI. Tailoring of rehabilitation to address individual needs for home-dwelling persons with TBI in the chronic phase represents an important area of future research.

## 1. Introduction

Traumatic brain injury (TBI) is a costly condition with long-lasting impact for many individuals [1,2,3]. Persons who suffer a TBI might experience a variety of consequences, including difficulties with physical, cognitive, emotional, behavioral, vocational, and social functioning. Many experience persistently reduced quality of life and restrictions in community participation [4,5,6,7,8,9]. Families are also affected and may have to adapt to a new life with their injured family member being dependent on their assistance and support [10,11,12,13,14]. It has been increasingly recognized that TBI is a chronic condition with multiple and interacting effects on health and wellbeing [15,16,17,18], as a significant proportion of patients continue to experience life-long difficulties and impaired functional status [8,19,20,21,22]. A challenge in rehabilitation after TBI is the heterogeneous nature of sequelae. Moreover, the patient’s specific difficulties interact with contextual and psychosocial factors [23,24]. Hence, many individuals are in need of long-term support from health care services. Evidence suggests that rehabilitation can be effective in reducing symptom burden and in improving participation and quality of life, and also for those who experience persisting symptoms [25,26,27,28]. However, evidence suggests that one-third of patients with chronic TBI have unmet needs related to cognitive, emotional, and vocational functioning [29], and that certain symptoms, such as neuropsychiatric sequelae, might often be overlooked in rehabilitation [30]. 

Rehabilitation efforts have become increasingly focused on enhancing patient involvement [31], and person-centered rehabilitation has been shown to have positive effects on occupational performance and rehabilitation satisfaction [32]. Goal-oriented rehabilitation with patient involvement is considered a key approach to rehabilitation [33,34], and has been shown to increase patient satisfaction and adherence [35], as well as improve self-efficacy, health-related quality of life and emotional status. There is, however, a need for more methodologically rigorous studies involving the use of individualized and specific treatment goals [36]. Although some studies have demonstrated the utility of a goal-oriented approach in tailoring rehabilitation efforts to the heterogeneous functional difficulties due to persistent TBI symptoms [37,38], more high-quality studies are needed on the effect of such approaches in the chronic phase of TBI. 

Although goal-oriented rehabilitation seems promising in chronic TBI, there might also be individual differences in the suitability of the approach. Many advocate that a high level of patient involvement is necessary in goal-oriented rehabilitation [34,39,40,41], and that patients with cognitive impairments are susceptible to being less involved in goal setting [42]. Cognitive impairment might, thus, lead to difficulties both with setting goals and with achieving them and should be explored when evaluating goal attainment [43]. Impaired self-awareness might be a particular challenge for patients with TBI, potentially influencing goal setting and engagement in rehabilitation [44]. Some studies have identified fatigue and emotional difficulties as potential barriers to early goal-oriented rehabilitation [45]. In addition, individual factors such as self-efficacy, tenacity, and motivation have further been identified as potential moderators of goal attainment [43,46,47]. To our knowledge, a systematic investigation of the degree to which cognitive impairment, emotional distress, demographic factors (i.e., age, gender, education), and/or injury-related variables predict goal attainment in the chronic phase of TBI has not yet been explored.

Despite the focus on goal-oriented rehabilitation over the past decades, conceptual terms vary, theoretical frameworks are often lacking [48,49], and there is a need to describe goal attainment [40,50], as goal attainment is rarely reported [51]. The SMART goal approach is frequently applied, i.e., setting goals that are Specific, Measurable, Achievable, Relevant, and Timed. Furthermore, the use of goal attainment scaling (GAS) [52] to measure goal attainment seems to be the best available alternative [53]. GAS is a systematic scoring of individualized goals in specific areas, which allows comparison of goal attainment across individualized goals and patients. GAS has been shown to be reliable, valid, and to have satisfactory responsiveness, as well as being sensitive to change [54]. Recently, Trevena-Peters, McKay [55] published results from a randomized controlled trial (RCT) supporting the effectiveness of an intervention to improve activities of daily living during post-traumatic amnesia, providing detailed results from GAS. A feasibility study of a project-based intervention for acquired brain injuries also detailed goal attainment results [56]. However, the studies neither provided information on the attainability of goals in distinct domains, nor did they investigate predictors of goal attainment.

The current study is modeled after a goal-oriented, home-based rehabilitation program shown to be effective in improving TBI-specific problem areas nominated by participants and which was shown to be highly acceptable for both patients and family members [57]. The current study represents an expansion and development of this approach in a different cultural setting (i.e., Norway), in a civilian sample, and with more severe injuries. The design was expanded by including SMART goals and GAS scoring within a randomized controlled trial, resulting in the combination of an individually targeted and standardized intervention approach. In addition to reporting group-based outcomes on standardized measures in the RCT, the design allows for exploration of the functional domains where individuals with TBI report a need for rehabilitation efforts. It also allows description of the degree to which setting individualized goals within the individual problem areas results in positive goal attainment. The study thus addresses several of the weaknesses in the current literature that have been noted above.

### Aims

The primary aim was to describe goal attainment in persons with persistent symptoms of TBI in the chronic phase. We hypothesized that participants would achieve goal attainment at expected levels. A second aim was to explore the functional domains of SMART goals established in the chronic phase and to determine whether goal attainment varied according to functional domains. We hypothesized that SMART goals would be related to physical/somatic, cognitive, emotional, and social problem areas typically seen in the chronic phase of TBI, and that goal attainment was achievable in all functional domains. Thirdly, we explored variables that might be associated with goal attainment, such as age, injury severity, and cognitive and emotional functioning. The existing literature does not give reason for a strong hypothesis regarding this aim; hence, this approach was considered exploratory in nature.

## 2. Materials and Methods

### 2.1. Participants

Participants were recruited from a two-group RCT conducted in Oslo, Norway. A detailed description of the study design is provided elsewhere [58]. Recruitment took place between June 2018 and December 2020. Between-group results of this trial will be published pending completion of 12 months follow-up assessments. Eligible participants were invited by letter, screened by phone, and, if eligible, invited to a baseline assessment at Oslo University Hospital (OUH). A family member was also invited if possible. Eligibility criteria were patients aged 18–72, with a TBI diagnosis with intracranial abnormalities verified by either computed tomography or magnetic resonance imaging. The participants had to be ≥16 years old at the time of injury, at least two years post-injury, and be living at home. Furthermore, they had to report ongoing TBI-related problems and/or reduced physical and mental health and/or difficulties with participation in their everyday life. Exclusion criteria were severe progressive neurologic or severe psychiatric disorders (including active substance abuse and violence), inability to provide informed consent, inability to participate in a goal-setting process, or insufficient fluency in Norwegian. After baseline assessment, participants were randomized 1:1 to either the control group or the intervention group by an independent researcher using a randomly generated number sequence. Participants in the control group received treatment as usual but no additional study-based treatment. Only patients randomized to the intervention group established SMART goals with subsequent GAS; hence, only results from the intervention group are reported in the current paper (*n* = 60). 

### 2.2. Intervention

The intervention group received a home-based intervention consisting of eight contacts over a 4-month period. Initially, six home visits and two telephone calls were carried out. Due to the Covid-19 pandemic, some patients were followed up by phone only during the initial Norwegian lockdown in March–May 2020. A pragmatic solution was adapted to continue recruitment during the pandemic, and most participants included from May to December 2020 (*n* = 17) were offered one to two home visits (first, ±last), while six to seven meetings were conducted by videoconference or telephone. Figure 1 displays an overview of the intervention sessions. Four therapists delivered the intervention: a medical doctor, a psychologist, a physiotherapist, and a neuropsychologist, all four with TBI rehabilitation expertise. Each participant was followed up by the same therapist throughout the intervention. 

The intervention was manualized and based on the study by Winter et al. [57]. It contained three phases: (1) identification of target problem areas, (2) establishment of SMART goals and GAS for the selected target problems, and (3) development of an Action Plan consisting of strategies to achieve the goal. Figure 2 displays an example of an action plan. Goals were established through brainstorming between the patient, therapist, and family member, and included identification of needs for support, barriers to change, and current adaptive strategies to be built upon. There was no upper limit on the number of SMART goals for each patient, but new goals were not established after session 5. The process of establishing SMART goals, GAS, and Action plans was based on recommendations for collaborative goal setting from several authors [50,59,60]. Patients were presented with visual and verbal information about the SMART approach to goal setting, and the SMART approach was applied in a flexible manner to increase patient involvement. Specific and written strategies to be employed to reach the SMART goals were established, based on collaborative interactions between participants, family members, and therapists. Therapists suggested a range of therapeutic strategies based on the current evidence base for the specific target problem area, and a list of common strategies was built up throughout the study related to recurring functional areas of SMART goals. Therapists reviewed and updated these strategies, and specific interventions were adopted to the individual needs of each patient. For details, see study protocol [58]. Team meetings were held on a regular basis, ensuring calibration of manual adherence across therapists. Ten percent of the sessions were observed by a senior professional with TBI expertise to evaluate treatment fidelity. 

### 2.3. Outcomes

#### 2.3.1. Goal Attainment Scaling

The main outcome measure in this study was goal attainment as measured by GAS scores, where five levels of goal attainment was agreed upon and established for each goal. GAS is, thus, subjective for each individual and goal specific. The expected level of goal attainment (scored as 0) was recorded, as well as two levels below the expected level (−1, −2; with baseline level being one of these) and two levels above the expected level (+1, +2). Baseline levels were set to −2 in cases where deterioration was impossible, and otherwise set to −1. Baseline levels were applied to evaluate change from the time at which the goal was set to GAS scoring at the last intervention session (session 8). To enhance precision, GAS levels were defined as specifically as possible, e.g., using percentages or number of days within the past week, as recommended by Malec [60]. Figure 2 displays an action plan example. At session 8, patient-reported goal attainment was registered, i.e., the patient’s own evaluation of their current goal level. In cases of reduced awareness or other factors influencing the patient reporting of goal attainment, therapist and family members interacted with the patient to establish consensus. 

Descriptive data are provided to depict the number of goals with goal attainment at the expected level or above, as well as goals with less than expected levels of attainment. As baseline GAS varied between −2 and −1, change scores were provided to describe goal attainment. GAS change scores were calculated as the difference between baseline and session 8 scores, and could, thus, vary between −1 (deterioration) and +4 (maximum improvement). A mean GAS score per participant was calculated by adding the raw change score for each goal and dividing the score on the number of goals for the specific individual.

#### 2.3.2. SMART Goal Categorization

To describe the functional domains covered by SMART goals, goals were categorized by two independent researchers (authors I.M.H.B. and S.L.H.) who identified goal themes based on the wording of each SMART goal. The categories were established earlier in the study to classify the target problem areas nominated by patients and family members, based on procedures described by Winter, Moriarty [61] and the International Classification of Functioning (ICF). See Borgen, Kleffelgaard [62] for an overview of this categorization of target outcomes. Twenty-four categories were established, which covered four overarching domains: cognitive, physical/somatic, emotional, and participation/social functioning. There was full agreement on categorization for 92% of the goals, and disagreements were resolved by consensus in the research group.

#### 2.3.3. Exploring Variables Associated with Goal Attainment 

Indicators of goal attainment were chosen within the domains of demographic variables, injury characteristics, intervention-related factors, cognitive functioning, global outcome, and self-reported symptoms. The data included in this analysis were collected at our outpatient clinic by members of the research team during the baseline assessment before randomization. Demographic data, i.e., age, work status (work percentage), and years of education was collected at baseline. Injury-related factors (i.e., injury severity, time since injury, and cause of injury) were retrieved from medical records. Injury severity was classified based on the lowest unsedated Glasgow Coma Scale (GCS) score the first 24 h after injury. GCS scores 3–8 were classified as severe TBI, 9–12 as moderate, and 13–15 as mild TBI [63]. Intervention-related factors included whether a family member participated and treatment expectation, the latter measured at session 1 and 3 by asking participants to rate their expectation that the intervention would be useful for them on a Likert scale from 1–10 (not at all to a very high degree). See Table 1 for an overview of standardized measures of global functioning, cognition, and self-reported symptoms, and their scorings [64,65,66,67,68,69,70,71,72,73]. 

### 2.4. Statistical Methods

All statistical tests were conducted in SPSS, version 26. Descriptions of patients and categorization of goals, as well as within-group changes in goal attainment from session 1 to session 8 are provided with descriptive statistics. Goal attainment per goal was not normally distributed, and Kruskal–Wallis H test was chosen to explore differences in goal attainment between domains. Distribution of GAS scores was assessed by visual inspection of QQ-plots. 

To determine indicators of GAS score at session 8, two analytical approaches were performed using multiple hierarchical linear regression analyses. In the first approach, variables based on theoretical, empirical, and clinical experience (“expert model”) were included in a hierarchical multiple regression analysis to compare models with or without controlling for baseline scores. Differences between the models were assessed with change in the explained variance (ΔR^2^) and whether this change was significant. In the second explorative approach all variables associated (*p* < 0.20) with GAS at session 8 from univariate regression analyses were included (“explorative model”), also controlling for baseline GAS levels in a block-wise approach. The chosen explorative variables are outlined above. One factor from each domain was chosen to avoid multicollinearity. Further, multicollinearity among exploratory variables was checked using Pearson correlation coefficient (r) or Spearman’s rho (ρ) of 0.7 as a cut off. The results from linear regression analyses are reported by regression coefficient (β) with 95% confidence interval (CI) and explained variance (R^2^). Changes in explained variance between the steps (ΔR^2^) and the significance levels are provided. Missing values of exploratory variables were 5% missing for cause of injury and 6% missing for injury severity. These data were multiple imputed under the assumption of missing at random. All available data were used to generate 15 imputed datasets. The results from each imputed dataset were combined to present single estimates.

### 2.5. Ethics

The study was approved by the Data Protection Office at OUH (2017/10390). The trial was registered at ClinicalTrials.gov, NCT03545594.

## 3. Results

### 3.1. Participants

Sixty participants were randomized to the intervention group. One withdrew after session 2 due to personal reasons, while the 59 remaining participants completed the intervention (session 8) and are included in the analysis. Thirty-nine (66%) had a participating family member, of whom 28 (72%) were spouses or domestic partners, 6 (15%) were parents, and 5 (13%) were other family members, such as siblings. Patient characteristics are reported in Table 2. In total, 56 (94%) participants participated in all 8 sessions, while 3 completed 7 sessions. Average length of intervention was 124 days (SD = 11.32; ~4 months). 

### 3.2. SMART Goals

In total, 151 unique SMART goals were established and rated at session 8, with a mean of 2.61 (SD = 0.72, range: 1–4) per participant. 

#### 3.2.1. Goal Attainment

At session 8, 41 (27%) goals were scored at expected levels of goal attainment (score 0), 55 (36%) goals were scored a little better than expected (score +1) and 42 (28%) goals were scored much better than expected (+2). Only 11 (7%) goals were scored a little worse than expected (−1), and 2 (1%) goals were scored as much worse than expected (−2) at session 8.

The median overall GAS change score was 2 (range: −1.0–4.0). At session 8, 141 (93.3%) of the goals showed positive goal attainment (i.e., change scores 1–4), while 1 (0.7%) goal was with a worse goal attainment than at baseline, and 9 (6.0%) goals were scored with no change from baseline. The mean raw GAS change score per participant (*n* = 59) was 2.22 (SD = 0.91), and mean improvement per participant ranged from 0.5 to 4.0, i.e., all participants improved on at least one of their goals. The mean GAS change score at the individual level is visualized in Figure 3.

#### 3.2.2. SMART Goal Domains and Categories

Table 3 displays the 151 SMART goals sorted by domains and sub-categories, with corresponding goal attainment. Table 3 also demonstrates that the SMART goals were classified within the same functional domains as target outcomes, confirming that goals adhered to problem areas initially reported by patients. The three most frequent SMART goal categories were related to reduced capacity and fatigue, memory difficulties, and sleep problems. Most goals were set related to physical/somatic functioning, especially regarding fatigue and sleep. Examples of such goals were “prevent episodes of fatigue >6 (VAS) during the week” and “maintain a circadian rhythm and get up at a fixed time”. Within the domain of cognitive functioning most goals were related to memory and cognitive executive functioning and included goals such as “establish routines to ensure finding my belongings” and “get started on everyday tasks and stop postponing things”. Goals regarding emotional functioning were most often related to anxiety and irritability and included goals such as “be less bothered by worrisome thoughts when going to bed” and “prevent and deal with episodes of irritability/anger in a calm manner”. Within the social domain, goals were most frequently related to social communication difficulties and included goals such as “contribute to a more open and positive family communication” and “manage to stop losing track and veering off-topic during conversations”. A Kruskal–Wallis H test was run to determine if there were differences in GAS change scores across the four goal domains, i.e., cognitive (*n* = 38), physical/somatic (*n* = 53), emotional (*n* = 35), and social (*n* = 25). Median GAS change scores were the same for all domains (2), with no significant differences between them (χ^2^(3) = 2.674, *p* = 0.445). 

#### 3.2.3. Indicators of Goal Attainment

The “expert model” included age, gender, injury severity, and total RPQ-score. The model showed an R^2^ = 0.128, F (5, 49) = 1.439, *p* = 0.227. Controlling for baseline levels gave an R^2^ change of 0.055 and a non-significant F change (*p* = 0.085).

As the model showed low predictive value, i.e., only predicted 12.8% of the total variance, univariate regression models were run to determine which explanatory variables should be included in the exploratory model. Results are presented in Table 4. 

The final exploratory model of factors with a significance level <0.2 thus included gender, anxiety symptoms, self-reported executive function (BRIEF-A GEC t-score), and treatment expectation at session 3 and GAS baseline levels. This model showed R^2^ of 0.322, F (5, 52) = 4.854, *p* = 0.001. The R^2^ change was 0.116, F change significance was *p* = 0.005, i.e., the adjusted model showed complete case (Table 4), and imputed models (data not shown) showed similar results.

## 4. Discussion

This study aimed at describing goal attainment in patients receiving an individually tailored, home-based rehabilitation intervention and at describing goal attainment in different goal domains. We also explored indicators of goal attainment at the final session.

Goal attainment was very high. All participants had a positive total goal attainment change score, which means that all participants improved on at least one of their goals. The high levels of goal attainment found across patients with different injury severity, time since injury, current level of functioning, and different goal domains indicated that the intervention format is well suited for many individuals in the chronic phase of TBI. We believe that the high level of patient involvement in this study might have resulted in the high goal attainment seen, as suggested in the literature [74]. Additionally, setting goals and GAS has been shown to be effective in and of itself [75], which may have contributed to the results. Goals were categorized as related to either cognitive, physical/somatic, emotional, or social functioning. The level of goal attainment was equal across goal domains, which implies that the intervention was sufficiently tailored to allow participants to work effectively on a broad range of issues. 

During baseline assessment in the RCT, patients and family members nominated target problem areas relating to TBI. A previously published paper [62] describes domains and categories of these problem areas. The problem areas reported at baseline were highly similar to the SMART goal areas reported in the current paper. A few problem areas reported at baseline were, however, not developed into SMART goals, i.e., visuospatial difficulties, reduced processing speed, difficulties with sensations, and difficulties with natural functions. Furthermore, some goal areas were not frequently established, such as goals related to identity difficulties and behavioral dysregulation. This may suggest that some problem areas are less easy to translate to SMART goals. If this was the result of difficulties in operationalizing abstract goal themes when applying GAS, this implies some limitation to the use of GAS. However, it might also be that abstract themes such as impaired self-awareness and identity difficulties were addressed while working on more concrete, everyday activities nominated by the patients, e.g., increased social activity. 

The initial investigation of indicators of goal attainment based on theoretical, empirical, and clinical perspectives, yielded a low predictive model explaining only 12.8% of the total variance of goal attainment in this sample. As the knowledge base about predictors of goal attainment is scarce, an exploratory approach was warranted to generate new hypotheses for future work. This approach suggested that being female, having low levels of anxiety symptoms, experiencing good executive functioning as well as high rehabilitation expectations were related to positive goal attainment. This finding should be interpreted with caution as there is a risk of overestimating the association of single explanatory variables in univariate regression analyses, and future investigation is needed. Furthermore, it should be noted that although the exploratory model is significant, the explained variance is still modest (32.2%), which implies that there are factors associated with goal attainment that were not included in the current model.

The fact that both demographic factors, emotional symptoms, TBI-related deficits, and factors relating to the intervention itself may play a role in goal attainment is, however, not surprising but clinically very important. Rehabilitation is a complex, multifaceted process that involves many interacting factors, and the identification of active ingredients in rehabilitation interventions is notoriously difficult [76]. It is not surprising that individual factors may be associated with intervention outcomes. In our exploratory model, neither age, education level nor employment status predicted goal attainment. However, women displayed higher goal attainment. This finding needs replication. The literature on the influence of gender on outcome post-TBI is mixed [77]. Colantonio and colleagues [78] found that men reported larger difficulties than women in setting realistic goals, which might influence goal attainment. Other studies have suggested that women might have more intact executive functioning and better self-awareness post-TBI, but findings vary, and other authors have suggested that women show higher levels of self-awareness after TBI [79]. Hence, we do not currently have any strong hypothesis regarding this result. The finding might even be spurious, in that gender is a proxy for a third and unknown variable. Interestingly, no injury-related factors were predictive of goal attainment. This could suggest that at the chronic stage of TBI, factors such as injury severity and time since injury do not play an important role in who benefits from every-day-oriented goal-based rehabilitation approaches. This supports the findings by Cicerone and colleagues that individuals with ongoing TBI-related difficulties should be offered support and may also benefit in the chronic stage, even years after the injury [25,26]. Additionally, self-reported executive dysfunction was shown to be detrimental to goal attainment, while performance-based cognitive impairments were not predictive of goal attainment. Thus, this only partly supports previous findings that cognitive impairment may hinder setting and achieving goals [42,43]. Despite previous findings that fatigue and emotional difficulties may be barriers to early goal-oriented rehabilitation in patients with stroke [80], only anxiety levels significantly predicted goal attainment in the current study. It may be that initial levels of fatigue and depression are a larger barrier to benefiting from rehabilitation during early recovery and are more addressable as the target of SMART goals later on. However, anxiety symptoms were shown to influence goal attainment. Anxiety levels are known to influence outcome post-TBI, although the directionality of this influence is disputed [81]. One study by Curran and colleagues [82] suggested that individuals with high levels of anxiety displayed more negative coping skills, such as worry, self-blame, and wishful thinking, and to some degree less positive coping skills such as problem solving. Whether anxiety symptoms in themselves are detrimental to goal attainment, or whether anxiety is a proxy for a variable such as coping skills is uncertain, and this finding also needs replication.

The finding that a positive expectation that the treatment could be beneficial during the third but not during the first session was predictive of goal attainment, was highly interesting. The finding may suggest that patient expectations are essential for goal attainment. However, as the wording of this question was the degree to which the participant expected that they would benefit from participating in the program, and that this belief was only predictive after participating in two or more sessions (and not at the very first session), it is likely that their response was influenced by their perceived level of therapeutic alliance. Although therapeutic alliance has received most attention in the field of psychotherapy, it has also been recognized as an important factor in brain injury rehabilitation (see [83] for a discussion). However, positive expectations might also be related to factors not measured in the current study. For example, the level of self-awareness may influence therapeutic alliance [44]. It may also be that expectations of change were influenced by the level of participant self-efficacy caused by the experienced improvement or lack thereof during the first three sessions. Self-efficacy, tenacity, and motivation have been previously shown to be predictive of goal attainment [34,41,44,45]. Future investigations should include measures of both therapeutic alliance, self-efficacy, and self-awareness in addition to change motivation to provide a clearer understanding of this interesting finding. The finding also indicated that treatment expectations should be discussed with patients early on in treatment, as this may play a role in treatment outcome. In summary, despite being exploratory, the current analyses provide hypotheses for further investigation of factors associated with goal attainment. Such investigations might be highly important to ensure a better understanding of what helps and what hinders goal attainment in rehabilitation, which again could help improve outcomes and ensure necessary tailoring of interventions.

### Limitations

This work has some limitations that should be recognized. Firstly, the comparability of goal attainment across patients when delivering an individualized intervention is always uncertain, and although the intervention was manualized, the specific content was tailored to the individual patient. However, the individualized nature of the intervention is also thought to be a major strength, given the heterogeneous nature of long-term symptoms of TBI, and because it allows participants to define for themselves what areas are important for them to work on, further enhancing patient involvement. Secondly, the efficacy of this intervention has not yet been established. Although this study is based on a similar RCT, which did demonstrate significant between-group effects [55], effects have not yet been investigated in our sample pending final outcome assessments. This entails that we do not yet know whether the high level of goal attainment is accompanied by improved participation and quality of life, which are the primary outcome measures in the RCT. However, high goal attainment is an important positive finding regardless of group average changes on global outcome measures. Thirdly, the sample may not be representative of patients with TBI in general. Rather, the study included those who continue to experience TBI-related challenges in everyday life and who were motivated to participate in rehabilitation. Thus, the sample is considered representative of patients seen in specialized rehabilitation clinics. Further, GAS scoring has some limitations, i.e., there may be reliability issues in the establishment and scoring of GAS. For example, there is a risk of the development of different procedures by each therapist, and, as noted earlier, the scoring is deemed to be subjective in nature. In this study, GAS scoring was conducted by the therapists, as scoring by a blinded third party was not feasible. How to best compute GAS scores across goals and individuals is also disputed, which is the reason that GAS change scores were applied instead of t-scores, as these are controversial [84]. In addition, it is important to note that the problem categories used in the current paper were based on previous work by our research group using a data-driven approach. Different approaches could be applied that might have resulted in a somewhat different categorization of goals. There is currently no gold standard in taxonomies for goal categorization, although some suggestions have been made elsewhere [85,86]. The exploratory regression models were conducted to generate hypothesis for future research, and identified factors should not be considered as predictive of goal attainment without replication. 

## 5. Conclusions

This study provides a transparent look at a goal-oriented approach in delivering rehabilitation interventions in the chronic phase of brain injury. Goal attainment was high, and goals were related to a broad range of problem areas typically identified in the chronic phase of TBI. Further investigation is needed to make strong conclusions regarding indicators of goal attainment, but the current study suggests that both individual, injury-related, and therapeutic factors are at play. The findings have clinical utility for therapists working with acquired brain injuries in general and other conditions where an individualized approach to treatment is warranted. 

## Figures and Tables

**Figure 1 jcm-11-00958-f001:**
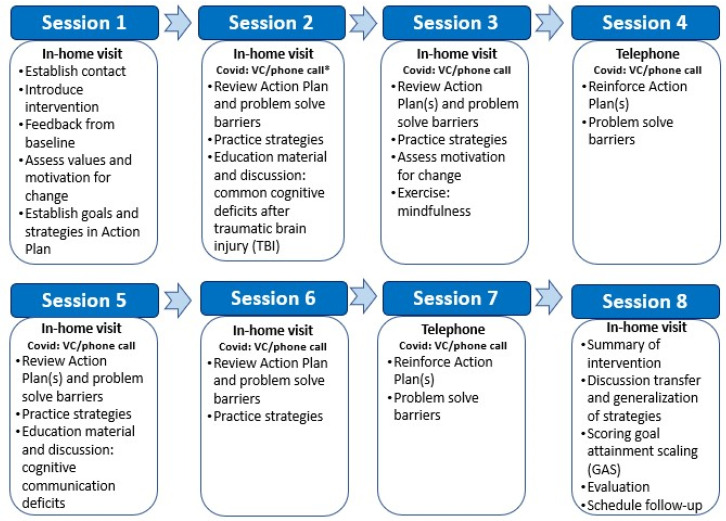
Overview of intervention sessions. * Delivery format was adjusted due to the Covid-19 pandemic, i.e., videoconference (VC) and phone calls replaced some home visits to reduce risk of infection.

**Figure 2 jcm-11-00958-f002:**
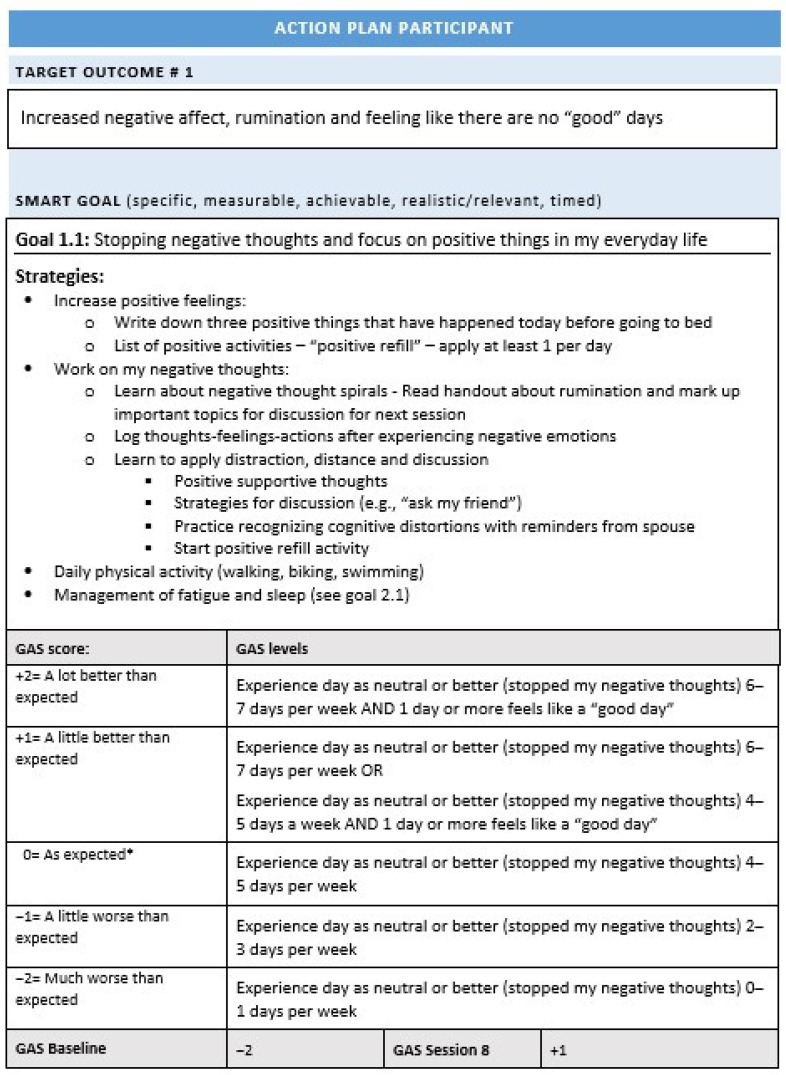
Action plan example with SMART goal, strategies, and GAS. * “As expected” here means the level you expect to accomplish before the program ends with a reasonable amount of effort.

**Figure 3 jcm-11-00958-f003:**
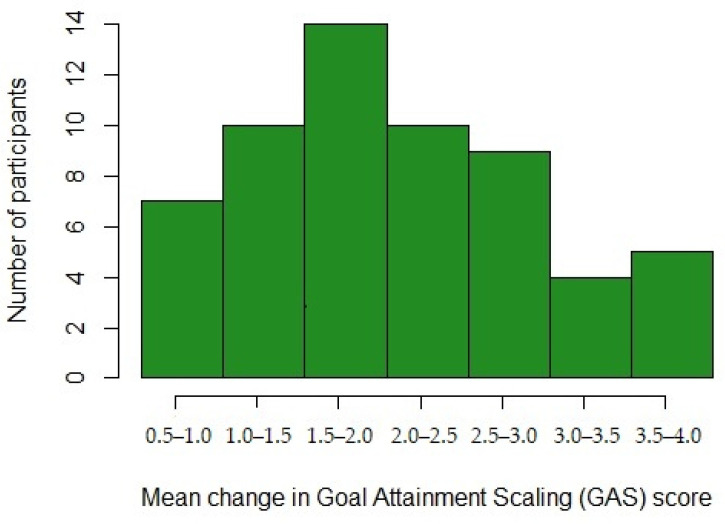
Mean individual change in GAS scores, across all 151 goals.

**Table 1 jcm-11-00958-t001:** Standardized outcomes and their applied scaling.

Assessment Domain	Measure Name	Score Used (Min.–Max.)
Global Outcome	GOSE [64]	Total score (3–8)
Cognitive functioning		
Verbal and visual abstraction/reasoning		
	Similarities and Matrices, WAIS-IV [65]	A dichotomized impairment variable was established, where impairment was defined as at least two test results being ≤1.5 standard deviation below the normative mean (no/yes) [66,67]
Verbal attention and working memory	Digit Span, WAIS-IV [65]	
Verbal learning and memory	CVLT-II [68]	
Processing speed, mental flexibility, and inhibition	Trail Making Tests 1–5 and Color Word Interference Tests 1–4, D-KEFS [69]	
Self-reported symptoms		
Post-concussive symptoms	RPQ [70]	Total score (0–64)
Fatigue	RPQ item [70]	Item score (0, 2–4)
Depressive symptoms	PHQ-9 [71]	Total score (0–27)
Anxiety-related symptoms	GAD-7 [72]	Total score (0–21)
Overall psychiatric distress	PHQ-9 [71] and GAD-7 [72]	Score of ≥10 on either scale (no/yes) [71,72]
Self-reported executive dysfunction	BRIEF-A [73]	Global Executive Composite t-score (0–100)

BRIEF-A = The Behavioral Rating of Executive Functions—Adult version, CVLT-II = California Verbal Learning Test-II, D-KEFS = Delis–Kaplan Executive Functioning Systems, GAD-7 = Generalized Anxiety Disorder 7-item, GOSE = Glasgow Outcome Scale Extended, PHQ-9 = Patient Health Questionnaire 9-item, RPQ = Rivermead Post-Concussion Symptoms Questionnaire, WAIS-IV = Weschler Adult Intelligence Scale IV.

**Table 2 jcm-11-00958-t002:** Patient characteristics.

Characteristics		Mean (SD)/Median (Range)/*n*(%)
*Demographics*		
Age, y		43.12 (13.61)
Gender, male		43 (73%)
Education, y		12 (10–20)
Marital status	Single	21 (36%)
	Married/domestic partner	32 (54%)
	Other (widowed, divorced, separated)	6 (10%)
*Injury-related factors*		
Injury severity (GCS) *		8 (3–15)
	Mild	16 (27%)
	Moderate	9 (15%)
	Severe	30 (51%)
	NA	4 (7%)
Cause of injury **	Fall	17 (29%)
	Transport-related	24 (40%)
	Violence	4 (7%)
	Other ^†^	11 (19%)
	NA	3 (5%)
Time since injury ***, y		4 (2–23)
*Work participation*		
Work percentage		0 (0–100)
Work status	Works full-time	16 (27%)
	Works part-time	13 (22%)
	Disability/sick leave/retired	30 (51%)

* *n* = 55. ** *n* = 56. *** *n* = 58. ^†^: sports- and leisure activities. GCS = Glasgow Coma Scale, SD = standard deviation, y = years.

**Table 3 jcm-11-00958-t003:** SMART goal categories and goal attainment at final session.

	Below Expectation	At Expectation	Above Expectation	Total
**Domain**/Category (number of participants)	*n*	*n*	*n*	*n*
**Cognitive difficulties**	**4 (11%)**	**11 (29%)**	**23 (60%)**	**38 (100%)**
Attention difficulties (*n* = 5, 9%)	1	2	4	7
Memory difficulties (*n* = 15, 25%)	3	6	11	20
Language difficulties (*n* = 1, 2%)	0	0	1	1
Cognitive aspects of executive functioning (*n* = 10, 17%)	0	3	7	10
**Physical/somatic difficulties**	**5 (9%)**	**13 (25%)**	**35 (66%)**	**53 (100%)**
Reduced capacity and fatigue (*n* = 21, 36%)	2	7	13	22
Pain (*n* = 4, 7%)	0	0	4	4
Sleep difficulties (*n* = 11, 19%)	1	1	10	12
Difficulties with motor functions (*n* = 6, 10%)	0	5	3	8
Difficulties with dizziness and balance (*n* = 7, 12%)	2	0	5	7
**Emotional difficulties**	**2 (6%)**	**8 (23%)**	**25 (71%)**	**35 (100%)**
Emotion perception and regulation (*n* = 3, 5%)	0	0	3	3
Irritability (*n* = 9, 15%)	1	3	6	10
Anxiety (*n* = 9, 15%)	0	2	8	10
Depressive thoughts and feelings (*n* = 8, 14%)	0	2	6	8
Difficulties with coping with stress (*n* = 3, 5%)	1	1	1	3
Difficulties with identity, acceptance, and sense of self (*n* = 1, 2%)	0	0	1	1
**Social function and participation**	**2 (8%)**	**9 (36%)**	**14 (56%)**	**25 (100%)**
Behavioral dysregulation (*n*= 1, 2%)	0	0	1	1
Social communication difficulties (*n* = 10, 17%)	0	3	6	9
Reduced self-sufficiency (*n* = 4, 7%)	0	2	2	4
Reduced social participation (*n* = 4, 7%)	0	1	3	4
Lack of meaningful activities (*n* = 6, 10%)	2	3	2	7
**Total**	**13 (8.6%)**	**41 (27.2%)**	**97 (64.2%)**	**151**

Number of participants with goal within each category is given in the left column. Goal attainment levels at session 8 are given as “below expectation” (score −2 or −1), “at expectation” (score 0), and “above expectation” (score +1 and +2). The total number of goals per domain/category registered at each level of attainment are given in *n* (%).

**Table 4 jcm-11-00958-t004:** Univariate regression analyses of goal attainment at final session (*n* = 59).

Exploratory Variables	B	95% CI	Significance	R Square	Decision
Demographic factors					
Age	0.002	−0.013 to 0.016	0.826	0.001	Discard
*Gender*	*0.327*	*−0.102 to 0.757*	*0.133*	*0.039*	*Keep*
Education (in years)	−0.042	−0.125 to 0.040	0.306	0.018	Discard
Percentage work participation (%)	0.002	−0.002 to 0.007	0.272	0.021	Discard
Injury-related factors					
GCS score	0.013	−0.034 to 0.060	0.588	0.006	Discard
Cause of injury (fall)	0.039	−0.125 to 0.204	0.633	0.004	Discard
Months since injury	0.000	−0.003 to 0.004	0.886	0.000	Discard
Functional status/symptoms at baseline					
Global functioning (GOSE)	−0.001	−0.202 to 0.201	0.994	0.000	Discard
Neuropsychology—overall impairment	−0.237	−0.623 to 0.149	0.224	0.026	Discard
Self-reported symptoms at baseline					
Post-concussion symptoms (RPQ total score)	0.004	−0.012 to 0.021	0.606	0.005	Discard
Fatigue (RPQ item)	0.047	−0.093 to 0.187	0.506	0.008	Discard
Depression (PHQ-9 total score)	−0.010	−0.045 to 0.026	0.589	0.005	Discard
*Anxiety (GAD-7 total score)*	*−0.032*	*−0.078 to 0.014*	*0.173*	*0.032*	*Keep*
Psychiatric symptoms (PHQ-9 and/or GAD-7 ≥ 10)	−0.072	−0.483 to 0.339	0.726	0.002	Discard
*Executive dysfunction (BRIEF-A GEC t-score)*	*−0.023*	*−0.044 to −0.002*	*0.032*	*0.080*	*Keep*
Intervention factors					
Treatment expectation at session 1	0.027	−0.074 to 0.127	0.595	0.005	Discard
*Treatment expectation at session 3*	*0.125*	*0.019 to 0.230*	*0.022*	*0.090*	*Keep*
Family member participation	0.099	−0.312 to 0.509	0.633	0.004	Discard

*Italics display results at acceptable p-value (<0.20) to be carried forward.* BRIEF-A = Behavioral Rating Inventory of Executive Functioning-Adult version. GAD-7 = Generalized Anxiety Disorder 7-items, GCS = Glasgow Coma Scale, GEC = Global Executive Composite, GOSE = Glasgow Outcome Scale Extended, PHQ-9 = Patient Health Questionnaire 9-item, RPQ = Rivermead Post-Concussion Questionnaire.

## Data Availability

Data can be viewed at secure servers at Oslo University Hospital by contacting the corresponding author.

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
