# Peer review of "Goal Attainment in an Individually Tailored and Home-Based Intervention in the Chronic Phase after Traumatic Brain Injury"

_jcm, 2022, doi:10.3390/jcm11040958_

Round 1

Reviewer 1 Report

Thank you for the opportunity to review your manuscript.  I have made some specific comments by section. 

Introduction:

Could enhance by including some additional references to support statements:

Line 50-52 – rehabilitation for TBI.  Reference 23/24 only relates to cognitive rehab.  Suggest including evidence of programs targeting rehabilitation goals in community-rehab context beyond cognitive rehabilitation.  For example, Evans et al 2008 (systematic review) – effectiveness of community-based rehab programs for adults with TBI, including group rehabilitation after TBI – for example: https://doi.org/10.3109/09638288.2015.1111436.  There is also research that has used goal-directed, individualised approaches with people with TBI in outpatient rehab contexts, particularly in the occupational therapy literature.  Authors state the need to investigate utility and tailoring rehab efforts to heterogeneous functional difficulties – need to consider that there are previous studies that have looked at this as described above.

Suggest paragraph 3, when talking about barriers to goal setting being cognitive impairment, reference be made specifically to impact of self-awareness on goal setting and engagement in rehabilitation.

The links between the literature cited in the introduction/concluding statements at the end of introductory paragraphs and the aims need to be made clear.  For example, ‘there is however a need for more methodologically rigorous studies involving the use of individualized and specific treatment goals [34]. Herein, there is a specific need to investigate the utility of goal- oriented approach in tailoring rehabilitation efforts to the heterogeneous functional difficulties due to persistent TBI-symptoms to target patients’ unmet needs (does this study aim to do this?).  Also, there is a need to evaluate goal attainment [36, 45], as goal attainment is rarely reported [46] (consider if this this study evaluating or describing goal attainment).  The link between paragraph 3 of the introduction (factors impacting on goal attainment and engagement in rehabilitation) and aim 3 (exploring relationships between goal attainment and other variables), is clear.

The word ‘evaluation of goal attainment; suggests that this study is evaluating the effectiveness of the intervention for achieving goal attainment – however this aim will be determined when results of the RCT are reported (comparing the intervention group with control group outcomes), which is not the subject of this paper.  As this paper is describing GAS goal outcomes post program, suggest aim one is to ‘describe, rather than evaluate, goal attainment at program end’.

Materials and methods:

  • Although this study refers reader to a study design paper (53) and describes the intervention (as the intervention group data is reported in this paper) it would be ideal for the reader to have some brief information about what the control group received in the participant section to be able to understand the context of what is reported here in the context of the larger study.
  • Regarding the measurement of goal attainment. Line 182 states ‘patients were asked to evaluate their goal attainment and suggest their current level’.  Please make it clear in the manuscript whether there were any other means of objectively establishing goal attainment routinely, or whether the primary mode of establishing pre-post intervention goal attainment was via patient report.
  • I wonder if there is a better heading to describe section 2.3.3 Identification of indicators of goal attainment. Variables measured to explore association with goal attainment?
  • Be clear in your method as to how goals were classified. At line 430 in the discussion, it states ‘it is important to note that the problem categories used in the current paper were based on previous work by our research group using a data-driven approach’ – this needs to be clearly described in the method (i.e. were goal statements open coded using qualitative approach)?
  • RE: description of variables (section 2.3.3) – be specific. work status (?what was collected); education (years education? Level – provide specific information). Injury severity – what was collected (duration PTA? Initial GCS?). GCS is reported in table 2 – be specific (was this initial GCS score?).  Also, how did you categorise mild/mod/severe (provide reference for the injury severity classification you used for GCS).  The likert scale 1-10 about usefulness (what were the descriptors for 1-10 ?not useful at all – extremely useful?).
  • When (what stage in the study) and how were these outcome measures other than GAS administered and by who (independent assessor/treating therapist?).

Results

  • Table 2 – note that % for injury severity categories do not add up to 100%
  • What does ‘at the individual level’ mean at line 267. Not clear.  Is this needed and is this just reporting ‘the mean GAS change in raw score per participant’.  Perhaps if you are reporting this, be clear in the method how you handled the GAS data.  For example ?For each individual, a mean GAS change score was calculated by (describe what you did).  As it is possible to calculate a t-score for individuals and evaluate change, it is important to make it clear that you are reporting mean raw individual change scores.  
  • Figure 2 – do you mean for the x axis values to be sitting in the middle of each bar (i.e. so that the first horizontal bar indicates that 7 participants had a mean GAS change score of 0.5?
  • Table 3 – providing % along with numbers in this table would assist with helping the reader to understand more easily whether statements in the discussion like ‘the level of goal attainment was equal across goal domains’ is supported.

Discussion

  • As suggested earlier, use word ‘describe’ goal attainment (line 299).
  • Not all aims are reiterated in opening paragraph (include describe domains in which participant’s goals were set)
  • Section giving example goals would be best placed in the results section alongside goals in each domain rather than discussion (Examples of such goals were “prevent ep- 317 isodes of fatigue >6 (VAS) during the week” and “maintain a circadian rhythm and get up 318 at a fixed time”. Within the domain of cognitive functioning most goals were related to 319 memory and cognitive executive functioning and included goals such as “establish rou- 320 tines to ensure finding my belongings” and “get started on everyday tasks and stop post- 321 poning things”. Goals regarding emotional functioning were most often related to anxiety 322 and irritability and included goals such as “be less bothered by worrisome thoughts when 323 going to bed” and “prevent and deal with episodes of irritability/anger in a calm manner”. 324), Within the social domain goals were most frequently related to social communication dif- 325 ficulties and included goals such as “contribute to a more open and positive family com- 326 munication” and “manage to stop losing track and veering off-topic during conversa- 327 tions”.
  • A previously published paper [57] describes domains and categories of these problem areas. The problem areas reported at baseline were highly similar to the SMART goal areas reported in the current paper (strengthen by providing detail here to help reader understand how findings support previous findings and what is new finding). Also the statement at 337 ‘This may suggest that some problem areas are less easy to translate to SMART goals’ – was it the case in your study that patient’s identified these goals but they weren’t worked on because they couldn’t be made SMART?  How were these types of goals handled.  Or was it that your participants in your study just didn’t identify these goals.  This problem of translating meaningful, person centred goals into measurable, GAS goals has been a criticism of GAS.  This would be an interesting discussion topic to expand upon in terms of the experience of using GAS in your study.
  • In line 342 you state: This was reported by therapists in the current study, and by therapists in the study by Winter and colleagues [70]. If indeed there was a qualitative component to your study which captures these utility components of goal setting, this needs to ideally be reported in the results rather than additional results being raised in the discussion.
  • The wording ‘indicators of goal attainment’ at line 402 – suggest using factors associated with goal attainment.
  • This section of the discussion at lines 416-421 I found confusing - This entails 416 that although participants in the intervention group display high levels of attainment on 417 specific goals, we do not yet know whether goal attainment is associated with improved 418 global outcomes regarding e.g., participation and quality of life. However, high goal at- 419 tainment is an important positive finding regardless of group comparisons on more global 420 outcome measures.
  • The statement ‘GAS-scoring has some limitations, including that reliability in identifying goals, 425 establishing, and scoring GAS might be an issue’at line 425 is very broad – be specific. How is GAS unreliable in identifying and establishing goals?  GAS is not the method used to ‘identify’ goals/guide goal setting conversations, rather it is what is used to drill down and operationalise identified goal areas into measurable, scaled goals. If patient-reported performance was used to determine goal attainment, rather than objective rating by therapist (even if not a blinded assessor), this is a potential limitation, particularly in a sample with cognitive impairment and likely self-awareness impairment post TBI.

Author Response

Reviewer 1

Thank you for the opportunity to review your manuscript.  I have made some specific comments by section. 

Thank you for your useful feedback and for taking the time to review our article.

Introduction:

Point 1: Could enhance by including some additional references to support statements:

Line 50-52 – rehabilitation for TBI.  Reference 23/24 only relates to cognitive rehab.  Suggest including evidence of programs targeting rehabilitation goals in community-rehab context beyond cognitive rehabilitation.  For example, Evans et al 2008 (systematic review) – effectiveness of community-based rehab programs for adults with TBI, including group rehabilitation after TBI – for example: https://doi.org/10.3109/09638288.2015.1111436. 

Response 1: Thank you. We agree that they are very relevant, and we have added the suggested references to this statement (line 51-53).

Point 2: There is also research that has used goal-directed, ndividualized approaches with people with TBI in outpatient rehab contexts, particularly in the occupational therapy literature.  Authors state the need to investigate utility and tailoring rehab efforts to heterogeneous functional difficulties – need to consider that there are previous studies that have looked at this as described above.

Response 2: Thank you for this comment. We have amended the text to address the fact that some studies have demonstrated the utility of this approach previously, and that the specific need is for methodologically robust studies for chronic TBI (line 64-68):

Although some studies have demonstrated the utility of a goal-oriented approach in tailoring rehabilitation efforts to the heterogeneous functional difficulties due to persistent TBI-symptom [37, 38], more high-quality studies are needed on the effect of such approaches in the chronic phase of TBI.

Point 3: Suggest paragraph 3, when talking about barriers to goal setting being cognitive impairment, reference be made specifically to impact of self-awareness on goal setting and engagement in rehabilitation.

Response 3: We agree that self-awareness should be mentioned in the introduction. We have added the following sentence to line 75-76:

“Impaired self-awareness might be a particular challenge for patients with TBI, potentially influencing goal setting and engagement in rehabilitation [44].”

Point 4: The links between the literature cited in the introduction/concluding statements at the end of introductory paragraphs and the aims need to be made clear.  For example, ‘there is however a need for more methodologically rigorous studies involving the use of individualized and specific treatment goals [34]. Herein, there is a specific need to investigate the utility of goal- oriented approach in tailoring rehabilitation efforts to the heterogeneous functional difficulties due to persistent TBI-symptoms to target patients’ unmet needs (does this study aim to do this?).  Also, there is a need to evaluate goal attainment [36, 45], as goal attainment is rarely reported [46] (consider if this this study evaluating or describing goal attainment).  The link between paragraph 3 of the introduction (factors impacting on goal attainment and engagement in rehabilitation) and aim 3 (exploring relationships between goal attainment and other variables), is clear.

Response 4: Thank you for this comment. Regarding the first paragraph noted we have deleted the end of the sentence referring to unmet needs, as we do not use objective measures of unmet needs, although it is implicitly assumed that the reported problem areas represent areas of unmet treatment needs. The sentence now ends with «tailoring rehabilitation efforts to the heterogeneous functional difficulties due to persistent TBI-symptoms».

Regarding the wording related to goal evaluation, we agree with the comment here and below, and have altered “evaluate” to “describe” throughout the manuscript.

In an effort to tie the Introduction together with the aims in a better way, we have also introduced a short summary at the end of the Introduction, and before aims. We hope this is considered an improvement. The last part of this paragraph now reads (line 105-112):

“The design was expanded by including SMART goals and GAS scoring within a randomized controlled trial, resulting in the combination of an individually targeted and standardized intervention approach. In addition to reporting group-based outcomes on standardized measures in the RCT, the design allows for exploration of the functional domains where individuals with TBI report a need for rehabilitation efforts. It also allows description of the degree to which setting individualized goals within the individual problem areas results in positive goal attainment. The study thus addresses several of the weaknesses in the current literature that have been noted above.»

Point 5: The word ‘evaluation of goal attainment; suggests that this study is evaluating the effectiveness of the intervention for achieving goal attainment – however this aim will be determined when results of the RCT are reported (comparing the intervention group with control group outcomes), which is not the subject of this paper.  As this paper is describing GAS goal outcomes post program, suggest aim one is to ‘describe, rather than evaluate, goal attainment at program end’

Response 5: We agree and have changed the word “evaluate” to “describe” throughout the manuscript.

Materials and methods:

Point 6: Although this study refers reader to a study design paper (53) and describes the intervention (as the intervention group data is reported in this paper) it would be ideal for the reader to have some brief information about what the control group received in the participant section to be able to understand the context of what is reported here in the context of the larger study.

Response 6: We agree that this would provide more context for the reader. We have added the following sentence (line 143-144): “Participants in the control group received treatment as usual, but no additional study-based treatment.”

Point 7: Regarding the measurement of goal attainment. Line 182 states ‘patients were asked to evaluate their goal attainment and suggest their current level’.  Please make it clear in the manuscript whether there were any other means of objectively establishing goal attainment routinely, or whether the primary mode of establishing pre-post intervention goal attainment was via patient report.

Response 7: Thank you for this comment, which relates to the challenge of objectively measuring attainment of individual and subjective goals. As noted in the manuscript, we based GAS scores on the patients´ evaluation. However, most participants had a participating family member. The therapist that had worked with the patients took part in the final session where GAS scores were established, Thus, as noted, there was a consensus-based scoring in cases where either the family member or the therapist experienced that the patient report was not accurate. We acknowledge the subjective nature of this procedure. On the other hand, we do not know of objective ways to register the “true” attainment of subjective and individually experienced change. Note that we will also publish the RCT results, where objective and standardized outcome measures will be applied. We have added the following clarification to the text (line 196-197):

“At session 8, patient-reported goal attainment was registered, i.e., the patient’s own evaluation of their current goal level. In cases of reduced awareness or other factors influencing the patient reporting of goal attainment, therapist and family members interacted with the patient to establish consensus.”

Point 8: I wonder if there is a better heading to describe section 2.3.3 Identification of indicators of goal attainment. Variables measured to explore association with goal attainment?

Response 8: We have amended the heading to now be “Exploring variables associated with goal attainment.

Point 9: Be clear in your method as to how goals were classified. At line 430 in the discussion, it states ‘it is important to note that the problem categories used in the current paper were based on previous work by our research group using a data-driven approach’ – this needs to be clearly described in the method (i.e. were goal statements open coded using qualitative approach)?

Response 9: We have added a sentence to the methods section stating that we categorized goal domains based on the wording of each SMART goal (line 213-214). The full sentence now reads:

«To describe the functional domains covered by SMART goals, goals were categorized by two independent researchers (authors I. M. H. B. & S.L.H) who identified goal themes based on the wording of each SMART goal”.

Point 10: RE: description of variables (section 2.3.3) – be specific. work status (?what was collected); education (years education? Level – provide specific information). Injury severity – what was collected (duration PTA? Initial GCS?). GCS is reported in table 2 – be specific (was this initial GCS score?).  Also, how did you categorise mild/mod/severe (provide reference for the injury severity classification you used for GCS).  The likert scale 1-10 about usefulness (what were the descriptors for 1-10 ?not useful at all – extremely useful?).

Response 10: Thank you for making us aware that specifics were lacking. We have made the following amendments:

(Line 228-229): “Demographic data, i.e., age, work status (work percentage), and years of education was collected at baseline”

(Line 230-233): “Injury severity was classified based on the lowest unsedated Glasgow Coma Scale (GCS) score the first 24 hours after injury. GCS scores 3-8 was classified as severe TBI, 9-12 as moderate and 13-15 as mild TBI [63].”

(Line 236): “asking participants to rate their expectation that the intervention would be useful for them on a Likert-scale from 1-10 (not at all-to a very high degree)”

Point 11: When (what stage in the study) and how were these outcome measures other than GAS administered and by who (independent assessor/treating therapist?).Response 11: The indicators of goal attainment are based on data collected at baseline. As randomization to the intervention group was done after baseline, these data were collected by members of the research group, and blinding was not necessary. Patients were assessed at our outpatient clinic. The following sentence has been added to this section (line 226-228).:

The data included in this analysis were collected at our outpatient clinic by members of the research team during the baseline assessment which was performed before randomization.”

Results

Point 12: Table 2 – note that % for injury severity categories do not add up to 100%

Response 12: Thank you for identifying this error. We have amended the table and the numbers now add up

Point 13: What does ‘at the individual level’ mean at line 267. Not clear.  Is this needed and is this just reporting ‘the mean GAS change in raw score per participant’.  Perhaps if you are reporting this, be clear in the method how you handled the GAS data.  For example ?For each individual, a mean GAS change score was calculated by (describe what you did).  As it is possible to calculate a t-score for individuals and evaluate change, it is important to make it clear that you are reporting mean raw individual change scores.  

Response 13: We agree that this should be explained more thoroughly. We have added the following to the text:

Method section (line 208-210):

“A mean GAS score per participant was calculated by adding the raw change score for each goal and dividing the score on the number of goals for the specific individual.”

Results (line 297):

“At the individual level, tThe mean raw GAS change score per participant (n=59) was 2.22 (SD=0.91)”

Point 14: Figure 2 – do you mean for the x axis values to be sitting in the middle of each bar (i.e. so that the first horizontal bar indicates that 7 participants had a mean GAS change score of 0.5?

Response 14: Thank you for this comment. We interpreted the comment as related to Figure 3 and have proceeded thereafter. This is a histogram were change scores are counted in 0.5 increments, e.g., 7 participants had a mean change score between 0.5 and 1.0. To make this figure clearer for the reader, the x-axis now provides the range of each bar.

Point 15: Table 3 – providing % along with numbers in this table would assist with helping the reader to understand more easily whether statements in the discussion like ‘the level of goal attainment was equal across goal domains’ is supported.

Response 15: We agree. We have now added % to the domain level. At the category level we feel this would not really have added meaning due to low number of cases in several cells.

Discussion

Point 16: As suggested earlier, use word ‘describe’ goal attainment (line 299).

Response 16: As noted, this has been done throughout.

Point 17: Not all aims are reiterated in opening paragraph (include describe domains in which participant’s goals were set)

Response 17: This paragraph now starts with the following sentence (line 349-351):

“This study aimed at describing goal attainment in patients receiving an individually tailored, home-based rehabilitation intervention, and at describing goal attainment in different goal domains”

Point 18: Section giving example goals would be best placed in the results section alongside goals in each domain rather than discussion (Examples of such goals were “prevent ep- 317 isodes of fatigue >6 (VAS) during the week” and “maintain a circadian rhythm and get up 318 at a fixed time”. Within the domain of cognitive functioning most goals were related to 319 memory and cognitive executive functioning and included goals such as “establish rou- 320 tines to ensure finding my belongings” and “get started on everyday tasks and stop post- 321 poning things”. Goals regarding emotional functioning were most often related to anxiety 322 and irritability and included goals such as “be less bothered by worrisome thoughts when 323 going to bed” and “prevent and deal with episodes of irritability/anger in a calm manner”. 324), Within the social domain goals were most frequently related to social communication dif- 325 ficulties and included goals such as “contribute to a more open and positive family com- 326 munication” and “manage to stop losing track and veering off-topic during conversa- 327 tions”.

Response 18: We agree that this should indeed be reported as a result, and have moved the suggested sentences to the results section.

Point 19: A previously published paper [57] describes domains and categories of these problem areas. The problem areas reported at baseline were highly similar to the SMART goal areas reported in the current paper (strengthen by providing detail here to help reader understand how findings support previous findings and what is new finding).

Response 19: The categories we applied in this paper are the same that those who were established in previous work by our group (reference 62). Thus, the categorization as such does not provide new findings. However, the categories were established based on the reported target outcome areas. This paper thus confirms that at the group level, the SMART-goals that were set adhered well to the domains noted by participants at their baseline assessment where target outcomes were reported. We have added the following sentence to section 3.2.2 of the results (line 307-309).:

«Table 3 also demonstrates that the SMART goals were classified within the same functional domains as target outcomes, confirming that goals adhered to problem areas initially reported by patients.”

Point 20: Also the statement at 337 ‘This may suggest that some problem areas are less easy to translate to SMART goals’ – was it the case in your study that patient’s identified these goals but they weren’t worked on because they couldn’t be made SMART?  How were these types of goals handled.  Or was it that your participants in your study just didn’t identify these goals.  This problem of translating meaningful, person centred goals into measurable, GAS goals has been a criticism of GAS.  This would be an interesting discussion topic to expand upon in terms of the experience of using GAS in your study.

Response 20: It is, as you say, a well-known discussion in relation to GAS that the need to operationalize outcome may lead to narrowing of goal contents. In the current study, we always took the patients´ wishes as a starting point. For some, this meant identification of goals that could be directly linked to abstract themes. However, in several cases we chose to work on themes that are difficult to operationalize such as loss of meaning, in indirect ways, e.g., through establishing goals related to ruminations or more social activities. In these cases, this was based on the patient’s own wishes and wording of these themes, and we wished to report goal themes within the context of their own wording of SMART goals. We have added the following discussion to the suggested paragraph (line 387-391):

“If this was the result of difficulties in operationalizing abstract goal themes when applying GAS, this implies some limitation to the use of GAS. However, it might also be that abstract themes such as impaired self-awareness and identity difficulties were addressed while working on more concrete, everyday activities nominated by the patients, e.g., increased social activity.”

Point 21: In line 342 you state: This was reported by therapists in the current study, and by therapists in the study by Winter and colleagues [70]. If indeed there was a qualitative component to your study which captures these utility components of goal setting, this needs to ideally be reported in the results rather than additional results being raised in the discussion.

Response 21: After reviewing your comment, we realized that the reporting we were referring to was mainly anecdotal, based on oral feedback from therapists in our study and that of Winter and colleagues. We thus have removed this statement from the text.

Point 22: The wording ‘indicators of goal attainment’ at line 402 – suggest using factors associated with goal attainment.

Response 22: We have amended the text in accordance with your suggestion (line 472).

Point 23: This section of the discussion at lines 416-421 I found confusing - This entails 416 that although participants in the intervention group display high levels of attainment on 417 specific goals, we do not yet know whether goal attainment is associated with improved 418 global outcomes regarding e.g., participation and quality of life. However, high goal at- 419 tainment is an important positive finding regardless of group comparisons on more global 420 outcome measures.

Response 23: We understand that this sentence is unclear. Our intention was to comment on the fact that we have not yet published the RCT results and do not know whether the intervention group changed on the study´s primary outcome measures compared to the control group. We have adjusted the paragraph, and it now reads (line 486-492):

«This entails that we do not yet know whether the high level of goal attainment is accompanied by improved participation and quality of life which are the primary outcome measures in the RCT. However, high goal attainment is an important positive finding regardless of group average changes on global outcome measures.”

Point 24: The statement ‘GAS-scoring has some limitations, including that reliability in identifying goals, 425 establishing, and scoring GAS might be an issue’at line 425 is very broad – be specific. How is GAS unreliable in identifying and establishing goals?  GAS is not the method used to ‘identify’ goals/guide goal setting conversations, rather it is what is used to drill down and operationalise identified goal areas into measurable, scaled goals.

Response 24: Thank you for this comment. We agree that the sentence was unclear, and that GAS is not a means to identify the goals. The sentence has been amended and now reads (line 496-499):

«Further, GAS-scoring has some limitations, i.e., that there may be reliability issues in the establishment and scoring of GAS. For example, there is a risk of development of different procedures by each therapist, and as noted earlier, the scoring is deemed to be subjective in nature.”

Point 25: If patient-reported performance was used to determine goal attainment, rather than objective rating by therapist (even if not a blinded assessor), this is a potential limitation, particularly in a sample with cognitive impairment and likely self-awareness impairment post TBI.

Response 25: See our reply above, where we refer to the fact that the goal attainment scoring was done in a conversation during the last session where the patient and therapist were both present, along with participating family members when they were involved.

Reviewer 2 Report

The present article evaluated goal attainment in those with persistent symptoms of TBI using patient expectations as the point of reference. It is a well written and interesting article. The following points should be addressed:

  1. You conclude that ‘these results support high level of goal attainment in patients with TBI participating in home-based, goal-oriented rehabilitation program in the chronic phase’. When an intervention is applied, the general tendency is to compare it with an already established therapeutic approach or a sham approach. That is even more important when assessing subjective outcome measures (patients tend to exaggerate in favour of the intervention). Therefore, uncontrolled studies are not usually used to extract such conclusions, but rather to indicate the potential efficiency of an intervention and suggest a need for further evaluation using controlled-blinded designs. Please reformulate the relevant parts of the texts according to these recommendations.
  2. Please complement your statistical plan with information regarding multicollinearity testing in the case of nominal and ordinal variables.
  3. GAS scores are a subjective approximation of goal attainment. You mention that ‘Baseline levels were set to -2 in cases where deterioration was impossible, 178 and otherwise set to -1’. Please provide the rational of using baseline GAS scores (since goals are formed at baseline and their attainment is assessed at follow-up).
  4. Moreover, using the above-mentioned approach, I feel that GAS change scores are not a suitable way to assess your outcome. Change scores are absolute measures that do not account for baseline levels. If relative scores are not used, it is preferable to analyse your scores (change scores or final scores) adjusted for baseline scores in you analysis (e.g., you may resort to univariate general linear models using baseline scores as covariates).
  5. Finally, please pay attention to the number of analyses and comparisons. I suggest a correction for multiple testing is implemented.

Author Response

Reviewer 2 - Comments and Suggestions for Authors

The present article evaluated goal attainment in those with persistent symptoms of TBI using patient expectations as the point of reference. It is a well written and interesting article. The following points should be addressed:

Thank you for reading our manuscript and for your useful comments. We have consulted our statistician (co-author CB) to ensure the proper handling of your methodological remarks.

Point 1: You conclude that ‘these results support high level of goal attainment in patients with TBI participating in home-based, goal-oriented rehabilitation program in the chronic phase’. When an intervention is applied, the general tendency is to compare it with an already established therapeutic approach or a sham approach. That is even more important when assessing subjective outcome measures (patients tend to exaggerate in favour of the intervention). Therefore, uncontrolled studies are not usually used to extract such conclusions, but rather to indicate the potential efficiency of an intervention and suggest a need for further evaluation using controlled-blinded designs. Please reformulate the relevant parts of the texts according to these recommendations.

Response 1: This sentence is in the abstract. In the Discussion section we have been clear that final group comparison and efficacy evaluation of the RCT is pending. The current paper refers to the intervention group only, and as such does not involve a comparison, but exploration of goal attainment as such. We however agree that the sentence in the abstract could give the impression that there was a comparison involved. We have amended the last two sentences in the abstract(line 29-33), which now reads:

“These results indicate a potential for high level of goal attainment in the chronic phase of TBI. Tailoring of rehabilitation to address individual needs for home-dwelling persons with TBI in the chronic phase represents important area of future research.”

Point 2: Please complement your statistical plan with information regarding multicollinearity testing in the case of nominal and ordinal variables.

Response 2: Thank you for this comment. We have added the specification that we have included Spearmans rho to test for multicollinearity, line 260-261.

Point 3: GAS scores are a subjective approximation of goal attainment. You mention that ‘Baseline levels were set to -2 in cases where deterioration was impossible, 178 and otherwise set to -1’. Please provide the rational of using baseline GAS scores (since goals are formed at baseline and their attainment is assessed at follow-up).

Response 3: We only denote the baseline scoring for two purposes. 1) the baseline scores could be either fixed or flexible, and we thus want to explain to the reader that we have used the flexible approach. 2) since we used a flexible approach to baseline scoring, this meant that the actual change in each goal was dependent on the baseline level. We thus used the baseline levels to calculate a change score for each goal set by each participant, with the thinking that their relative change within one goal was more important than the endpoint. In the new analyses (see below), we also used the baseline levels to control for endpoint scores. We have added the following sentence to the text (line 191-192) to clarify:

“Baseline levels were applied to evaluate change from the time at which the goal was set to GAS scoring at the last intervention session (session 8).”

Point 4: Moreover, using the above-mentioned approach, I feel that GAS change scores are not a suitable way to assess your outcome. Change scores are absolute measures that do not account for baseline levels. If relative scores are not used, it is preferable to analyse your scores (change scores or final scores) adjusted for baseline scores in you analysis (e.g., you may resort to univariate general linear models using baseline scores as covariates).

Response 4: Thank you for this comment. We agree that other approaches than that conducted in our draft is viable for these type of analyses. To adhere to your advice, we decided to redo the analyses. As adding baseline scores as covariates accounts for baseline levels, we decided to use the final scores (GAS at session 8) as the endpoint. We have therefore updated the description of this approach in the methods section, line 248-264. We thus conducted both the expert model and the explorative model (both the univariate analyses and the final model) again and have reported the new results in Table 4 and on line 330-343. The discussion has been updated also, see line 399-451. As this new approach showed that two factors were substituted in the final exploratory model, we felt this confirms the exploratory nature of the analysis and have reiterated this point in the limitation section (line 509-511).

Point 5: Finally, please pay attention to the number of analyses and comparisons. I suggest a correction for multiple testing is implemented.

Response 5: We use the single regressions to identify factors for the exploratory model and are not weighting the p value of the single predictors. Explained variance and not statistical significance is applied to evaluate the models hence we have not adjusted for multiple comparisons.

Round 2

Reviewer 2 Report

Thank you for considering my suggestions